# The effect of antibiotic prescription in non-critically ill hospitalized patients with COVID-19: A Japanese inpatient database study

**Haruka Imai**[1,2,3,4], **Jun Suzuki**[2,3,4], **Tomoki Mizuno**[2,3], **Shota Takahashi**[3], **Hideya Itagaki**[5], **Makiko Yoshida**[2,3,4,6], **Shiro Endo**[2,3,4,5,6], **Eiichi N. Kodama**[1,7]*

**1** Department of Infectious Diseases, Graduate School of Medicine, Tohoku University, Sendai, Japan, **2** Division of Infectious Diseases, Tohoku Medical and Pharmaceutical University Hospital, Miyagi, Japan, **3** Department of Infection Prevention and Control, Tohoku Medical and Pharmaceutical University Hospital, Miyagi, Japan, **4** Division of Infectious Diseases and Infection Control, Faculty of Medicine, Tohoku Medical and Pharmaceutical University, Miyagi, Japan, **5** Division of Infectious Diseases and Infection Control, Department of Social and Community Medicine, Graduate School of Medicine, Tohoku Medical and Pharmaceutical University, Miyagi, Japan, **6** Division of the Crisis Management Network for Infectious Diseases, Tohoku Medical and Pharmaceutical University, Miyagi, Japan, **7** Division of Infectious Disease, International Research Institute of Disaster Science, Tohoku Medical Megabank Organization, Tohoku University, Sendai, Japan

* eiichi.kodama.e2@tohoku.ac.jp

## Abstract

### Background

Coronavirus disease 2019 (COVID-19) is an ongoing global pandemic. Bacterial coinfections with COVID-19 occur in 3.5% of COVID-19 cases, with a higher incidence in severe cases. Although antibiotics have been prescribed to treat non-critically ill patients with COVID-19, their effect on non-critically ill hospitalized patients with COVID-19 remains uncertain.

### Methods

We analyzed data from non-critically ill hospitalized patients with COVID-19 who were older than 18 years between January 1, 2020, and May 31, 2023. We performed propensity score matching analysis, evaluating in-hospital mortality with or without antibiotic prescription within 2 days of admission. Sensitivity analyses using inverse probability weighting and generalized estimating equation were also performed.

### Results

Eligible patients (n = 144,110) were divided into antibiotic prescription (n = 3,873) and control (n = 140,237) groups. One-to-one propensity score matching identified 3,861 pairs of patients who received antibiotic prescriptions within 2 days of admission. Following this, antibiotic prescription was associated with a decreased 28-day mortality rate (2.3% vs. 3.6%) and in-hospital mortality rate (4.0% vs. 5.0%) compared with the control group.

**Data availability statement:** Minimal data for this study can not be shared publicly, because the data was obtained from a third-party (Medical Data Vision). Data are available from Medical Data Vision (https://en.mdv.co.jp) contact, Yuki Santo, via email (santo_yuki@mdv.co.jp) for researchers who meet the criteria for access to confidential data. The dataset includes COVID-19 patient information from 1 January 2020 to 31 May 2023. The authors had no special access privileges and confirm other researchers will be able to obtain the same data and replicate the study results by following the Methods presented in the paper.

**Funding:** This work was supported by JSPS KAKENHI (grant number: 23K16288), received by Jun Suzuki, and AMED-CREST (JP22gm1610007), received by Eiichi N Kodama.

**Competing interests:** The authors have declared that no competing interests exist.

Conversely, antibiotics increased *Clostridioides difficile* infection (CDI) compared with the control group (0.6% vs. 0.1%). No statistical differences were observed between both groups regarding acute kidney injury (0.4% vs. 0.2%). Sensitivity analysis showed similar outcomes.

## Conclusions

This multicenter observational study in Japan showed that antibiotic prescriptions were associated with lower 28-day and in-hospital mortalities and an increased CDI risk in non-critically ill hospitalized patients with COVID-19.

## Introduction

Coronavirus disease 2019 (COVID-19), caused by severe acute respiratory syndrome coronavirus 2 (SARS-CoV-2), can progress from mild symptoms to acute respiratory distress syndrome with a high mortality rate, leading to a global pandemic. COVID-19 causes problems in both its acute and chronic phases, such as post-COVID-19 condition [1–4]. Thus, COVID-19 treatment strategies have been investigated in many countries to reduce its spread and associated mortality [5–8].

A possible benefit of antimicrobial use in COVID-19 patients is the possibility of reduced mortality [9]. Disadvantages include increased antimicrobial resistance [10,11], diarrhea, other gastrointestinal problems, and *Clostridioides difficile* infection [12].

Previous studies have shown that the rates of bacterial complications associated with COVID-19 are approximately 3.5%, with a high incidence in severe cases, necessitating antibiotic prescription [13]. Conversely, antibiotic usage in patients with COVID-19 was approximately 62% abroad [14]. In Japan, antimicrobials were prescribed to 9% of COVID-19 outpatients [15] and 16% of inpatients [16]. Thus, inappropriate antibiotics may be prescribed to patients with COVID-19. Antibiotic use increases the risk of *Clostridioides difficile* infection (CDI) and acute kidney injury (AKI) [17]. However, the beneficial and disadvantageous effects of antibiotics in non-critical COVID-19 inpatients remain unclear. Therefore, this study aimed to evaluate the effects of antibiotic prescriptions in non-critically ill patients with COVID-19 using a Japanese inpatient database.

## Materials and methods

### Database information

We utilized a Japanese multicenter inpatient database purchased from Medical Data Vision (MDV; Tokyo, Japan). MDV employs anonymized processed data (e.g., medical records and test results) provided by medical institutions for analysis and service development, ensuring strict confidentiality. During processing, personally identifiable information, such as date of birth and address, is obscured to prevent patient identification. This anonymized data is securely provided, for instance, by being downloaded from our servers or uploaded to the designated server [18].

As of the end of May 2023, the MDV database included outpatient and inpatient information for 43.83 million patients from approximately 400 medical institutions using the Diagnosis Procedure Combination (DPC) system [19]. The database contains anonymized information on diagnoses, patient characteristics, drug prescriptions, medical procedures, characteristics of medical institutions, and medical reimbursement costs [20].

## DPC data

DPC, a patient classification system initiated by the Ministry of Health, Labor, and Welfare (MHLW) in 2002, is associated with a flat rate per diem payment system. The system is accepted by over 1,700 large- and medium-sized hospitals in Japan, and the DPC database contains administrative claims and discharge summaries. Data elements include hospital-specific identifiers such as age, sex, primary diagnosis, admission comorbidities, post-admission complications, surgical procedures, anesthesia time, stay length, discharge status, total hospital charges, smoking status, obesity, level of consciousness, and activities of daily living. The attending physician recorded all patients' data upon discharge [20].

## Patient selection

We identified non-critically ill patients with COVID-19 between January 1, 2020, and May 31, 2023. Non-critically ill hospitalized patients were those who did not receive oxygen, high-flow therapy, mechanical ventilation support, or oral and intravenous steroids within 2 days of admission, as defined by MHLW [21]. Exclusion criteria were as follows: 1) under 18 years of age; 2) discharged within 2 days of admission for any reason; 3) pregnant patients; 4) oxygen therapy; 5) steroid administration; 6) mechanical ventilation; 7) nasal high-flow therapy; 8) history of kidney disease; 9) hemodialysis; 10) continuous hemodiafiltration; 11) oral vancomycin; 12) oral fidaxomicin; 13) oral metronidazole; and 14) intravenous metronidazole, all within 2 days of admission (Fig 1).

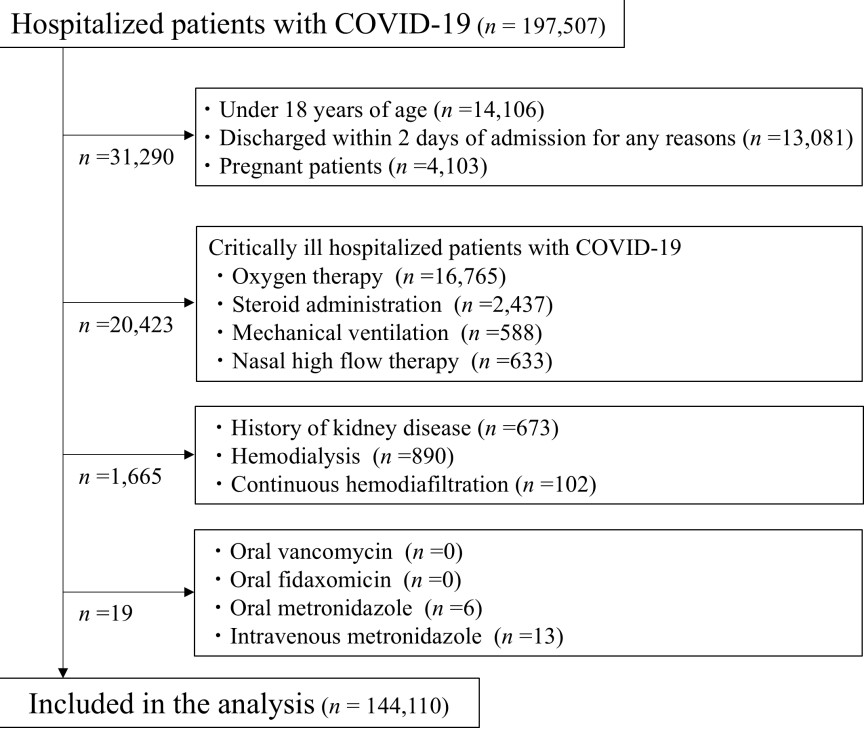

**Fig 1. Flow chart for patients inclusion.**

## Variables

The exposure of interest determined whether patients received antibiotics. The antibiotic prescription group included patients who received antibiotics within 2days of admission, whereas the control group included those who did not. The variables assessed included age, age distribution, sex, body mass index (BMI), smoking status, ambulance use, Charlson Comorbidity index (CCI) score, comorbidities at admission time, and consciousness level. Comorbidities at admission time were defined per the CCI score [22], such as myocardial infarction, congestive heart failure, peripheral vascular disease, cerebrovascular disease, dementia, chronic pulmonary disease, peptic ulcer, malignancy, metastatic solid tumor, mild liver disease, moderate or severe liver disease, hemiplegia or paraplegia, diabetes with or without chronic complications, and rheumatic disease.

Japan coma scale (JCS) score was recorded for all patients to assess their consciousness level on admission, correlating well with the Glasgow Coma Scale score [23]. The JCS scores were divided into four categories: 0 (alert), 1–3 (delirium), 10–30 (somnolence), and 100–300 (coma) [23].

We also investigated the following procedures performed within 2 days of admission: intravenous remdesivir, oral molnupiravir, inhaled ciclesonide, and anticoagulants. Anticoagulants included intravenous and intramuscular heparin and direct oral anticoagulants, such as apixaban, edoxaban, rivaroxaban, dabigatran, and oral warfarin.

## Definition of CDI and AKI

We attempted to detect CDI and AKI using ICD-10 disease codes, which were rarely recorded in this database. Definitions were based on therapeutic agents, as the database lacks laboratory data and bacterial culture results. CDI was defined as patients with COVID-19 treated with oral vancomycin, oral metronidazole, or intravenous metronidazole 3 days post-hospitalization. Oral fidaxomicin was not prescribed 3 days post-hospitalization and was excluded from the definition. AKI was defined as patients with COVID-19 treated with continuous hemodiafiltration and hemodialysis 3 days post-hospitalization.

## Outcome measures

The main outcomes were 28-day and in-hospital mortality rates, including CDI and AKI rates.

## Statistical analysis

We performed a one-to-one propensity score matching to adjust for differences in baseline characteristics between the antibiotic prescription and control groups. Estimating the propensity score, the probability of a patient receiving antibiotics was adjusted for potential confounders using the following characteristics: age, age distribution, sex, BMI, smoking status, ambulance use, CCI, comorbidities at admission time, consciousness level, and COVID-19 medications within 2 days of admission (intravenous remdesivir, oral molnupiravir, inhaled ciclesonide, and anticoagulants). C-statistic was calculated to evaluate goodness-of-fit. One-to-one matched analysis using nearest-neighbor matching was performed based on the propensity scores. A match occurred when a patient in the antibiotic prescription group had an estimated score within 0.2 standard deviations of a patient in the control group. We used absolute standardized mean differences to assess the balance between patient characteristics. Standardized differences within ±10% were considered negligible imbalances in baseline characteristics between the groups. Descriptive data were reported as numbers and percentages (for categorical variables) and medians and interquartile ranges (for continuous

variables). Descriptive statistics were assessed before and after propensity score matching. We also used risk differences to compare outcomes between the groups. Additionally, we performed sensitivity analyses using inverse probability of treatment weighting (IPTW) and generalized estimating equations (GEEs), using logistic regression analysis fitted with a GEE-adjusted propensity score to investigate the association between antibiotics and outcomes by estimating odds ratios (ORs) and confidence intervals (CIs). Propensity score matching, IPTW, and GEEs were performed using Stata/MP 17.0 (Stata Corp, College Station, TX, USA).

### IRB information

This study was approved by the Institutional Review Board of Tohoku Medical and Pharmaceutical University (approval number: 2023-025). The need for informed consent was waived because of the anonymous nature of the data.

## Results

Exactly 144,110 patients (73%) were finally included in the analysis (Fig 1, Table 1). Eligible patients were divided into the antibiotic prescription group (n = 3,873) and the control group (n = 140,237). One-to-one propensity score matching identified 3,861 pairs of patients who received antibiotic prescriptions within 2 days of admission. C-statistic indicated that the goodness-of-fit was 0.794 (95% CI, 0.786–0.803) in the propensity score model.

Baseline characteristics of the groups before and after propensity score matching are summarized in Table 1. Patient characteristics were well-balanced between the groups following propensity score matching with the standardized differences within ± 10%. The baseline characteristics after IPTW analysis are presented in Table 2, where patient characteristics were well-balanced between the groups, with standardized differences within ± 10%.

### Outcome measures

The outcomes of the antibiotic prescription and control groups before and after propensity score matching are shown in Table 3. Pre-propensity score matching, antibiotic use was associated with a decline in the 28-day mortality rate (2.4% in the antibiotic prescription vs. 3.1% in the control groups; p = 0.01). Conversely, antibiotics were associated with an increase in CDI (0.6% in the antibiotic prescription vs. 0.01% in the control groups; p < 0.01). No statistically significant differences were observed between the antibiotic prescription and control groups regarding hospital mortality (4.0% vs. 4.2%, respectively; p = 0.47) or AKI (0.4% vs. 0.03%, respectively; p = 0.23). Following propensity score matching, antibiotics were associated with a decrease in the 28-day mortality rate (2.3 in the antibiotic prescription vs. 3.6% in the control groups; p < 0.01) and in-hospital mortality rate (4.0 in the antibiotic prescription vs. 5.0% in the control groups; p = 0.03). Conversely, antibiotics were associated with an increase in CDI (0.6% in the antibiotic prescription group vs. 0.1% in the control group; p < 0.01). No significant differences were observed between both groups regarding AKI (0.4% in the antibiotic prescription vs. 0.2% in the control groups; p = 0.12). IPTW analysis estimates are shown in Table 4, and IPTW findings indicated similar outcomes to those observed post-propensity score matching.

The association between antibiotic prescription and GEE analysis outcomes for all included patients is shown in Table 5. The GEE results were 28-day mortality (OR, 0.7; 95% CI, 0.5–0.8), in-hospital mortality (OR, 0.8; 95% CI, 0.7–0.95), CDI (OR, 4.7; 95% CI, 2.5–8.6), and AKI (OR, 1.4; 95% CI, 0.8–2.7).

**Table 1. Patient baseline characteristics in the unmatched and propensity score-matched groups.**

| | Unmatched groups | | | Propensity score-matched groups | | |
|---|---|---|---|---|---|---|
| | Antibiotic pre-scribing group | Control group | Standardized difference | Antibiotic pre-scribing group | Control Group | Standardized difference |
| **Variables** | n = 3,873 | n = 140,237 | | n = 3,861 | n = 3,861 | |
| **Age median, (IQR) (years)** | 80 [66–88] | 68 [48–82] | 0.48 | 80 [66–88] | 80 [67–88] | −0.03 |
| **Sex (man), n (%)** | 2,027(52.3) | 76,480 (54.5) | 0.04 | 2,017 (52.2) | 1,980 (51.3) | −0.01 |
| **BMI (kg/m²), n (%)** | | | | | | |
| <18.5 | 1,573 (40.6) | 31,091 (22.2) | 0.41 | 1,566 (40.6) | 1,576 (40.8) | −0.01 |
| 18.5–25 | 1,750 (45.2) | 70,523 (50.3) | −0.10 | 1,746 (45.2) | 1,727 (44.7) | 0.01 |
| 25–30 | 429 (11.1) | 27,760 (19.8) | −0.24 | 429 (11.1) | 440 (11.4) | 0.02 |
| 30–35 | 89 (2.3) | 7,947 (5.7) | −0.17 | 89 (2.3) | 85 (2.2) | −0.02 |
| >35 | 32 (0.8) | 2,889 (2.1) | −0.10 | 31 (0.80) | 38 (0.98) | −0.01 |
| missing data | 754 (19.5) | 15,055 (10.7) | 0.25 | 748 (19.4) | 737 (19.1) | 0.00 |
| **Smoking, n (%)** | | | | | | |
| current or past smoker | 726 (18.8) | 37,056 (26.4) | −0.19 | 724 (18.8) | 715 (18.5) | −0.01 |
| non-smoker | 2,264 (58.5) | 81,704 (58.3) | 0.01 | 2,260 (58.5) | 2,276 (59.0) | 0.01 |
| missing data | 882 (22.8) | 21450 (15.3) | 0.19 | 877 (22.7) | 870 (22.5) | −0.01 |
| **Ambulance, n (%)** | 1,631 (42.1) | 38,583 (27.5) | 0.31 | 1,625 (42.1) | 1,660 (43.0) | −0.02 |
| **CCI median, (IQR)** | 5.0 [3.0–5.0] | 3.0 [1.0–5.0] | 0.47 | 5.0 [3.0–5.0] | 5.0 [3.0–5.0] | −0.04 |
| **Comorbidities, n (%)** | | | | | | |
| Myocardial infarction | 6 (0.15) | 90 (0.06) | 0.03 | 6 (0.16) | 10 (0.26) | −0.02 |
| Congestive heart failure | 24 (0.62) | 797 (0.57) | 0.01 | 24 (0.62) | 22 (0.57) | −0.02 |
| Peripheral vascular disease | 2 (0.05) | 61 (0.04) | 0.00 | 2 (0.05) | 3 (0.08) | 0.00 |
| Cerebrovascular disease | 31 (0.80) | 666 (0.47) | 0.04 | 31 (0.80) | 38 (0.98) | −0.02 |
| Dementia | 47 (1.2) | 680 (0.48) | 0.08 | 47 (1.2) | 34 (0.88) | 0.05 |
| Chronic pulmonary disease | 5 (0.13) | 334 (0.24) | −0.03 | 5 (0.13) | 5 (0.13) | 0.00 |
| Peptic ulcer | 16 (0.41) | 261 (0.19) | 0.04 | 16 (0.41) | 11 (0.28) | −0.01 |
| Malignancy | 41 (1.1) | 870 (0.62) | 0.05 | 41 (1.1) | 51 (1.3) | −0.04 |
| Metastatic solid tumor | 5 (0.13) | 175 (0.12) | 0.00 | 5 (0.13) | 5 (0.13) | 0.00 |
| Mild liver disease | 13 (0.34) | 206 (0.15) | 0.04 | 13 (0.34) | 15 (0.39) | −0.02 |
| Moderate or Severe liver disease | 1 (0.03) | 40 (0.03) | 0.00 | 1 (0.03) | 5 (0.13) | −0.01 |
| Hemiplegia or paraplegia | 3 (0.08) | 21 (0.01) | 0.03 | 3 (0.08) | 3 (0.08) | −0.01 |
| Diabetes with chronic complications | 43 (1.1) | 998 (0.71) | 0.04 | 43 (1.1) | 36 (0.93) | −0.03 |
| Diabetes without chronic complications | 34 (0.88) | 831 (0.59) | 0.03 | 34 (0.88) | 26 (0.67) | −0.03 |
| Rheumatic disease | 1 (0.03) | 40 (0.03) | 0.00 | 1 (0.03) | 1 (0.03) | −0.01 |
| **Consciousness levels, n (%)** | | | | | | |
| 0 (Alert) | 2,679 (69.2) | 120,188 (85.7) | −0.40 | 2,674 (69.3) | 2,635 (68.3) | 0.04 |
| 1–3 (Delirium) | 965 (24.9) | 16,248 (11.6) | 0.35 | 962 (24.9) | 964 (25.0) | −0.02 |
| 10–30 (Somnolence) | 179 (4.6) | 2,404 (1.7) | 0.17 | 175 (4.5) | 196 (5.1) | −0.04 |
| 100–300 (Coma) | 32 (0.83) | 1,133 (0.81) | 0.00 | 32 (0.83) | 43 (1.1) | 0.01 |
| **Interventions within 2 days of admission, n (%)** | | | | | | |
| Anticoagulation | 533 (13.8) | 2,249 (1.6) | 0.47 | 524 (13.6) | 432 (11.2) | 0.096 |
| Inhaled ciclesonide | 6 (0.15) | 20 (0.01) | 0.05 | 6 (0.16) | 6 (0.16) | 0.01 |
| Molnupiravir | 107 (2.8) | 522 (0.37) | 0.19 | 100 (2.6) | 112(2.9) | −0.04 |
| Remdesivir | 1,438 (37.1) | 5,211 (3.7) | 0.91 | 1,428 (37.0) | 1,443 (37.4) | 0.00 |

Abbreviations: IQR, interquartile range; BMI, body mass index; CCI, Charlson Comorbidity index.

**Table 2. Patient baseline characteristics in the IPTW analysis group.**

| | IPTW analysis group | | |
|---|---|---|---|
| | Antibiotic prescribing group | Control Group | Standardized difference |
| **Variables** | | | |
| **Mean age, (years)** | 65.46 | 64.60 | 0.04 |
| **Sex** | 0.50 | 0.54 | -0.09 |
| **BMI (kg/m²)** | | | |
| <18.5 | 0.25 | 0.23 | 0.05 |
| 18.5–25 | 0.50 | 0.50 | 0.00 |
| 25–30 | 0.18 | 0.20 | -0.04 |
| 30–35 | 0.05 | 0.06 | -0.01 |
| >35 | 0.01 | 0.02 | -0.05 |
| missing data | 0.13 | 0.11 | 0.05 |
| **Smoking** | | | |
| current or past smoker | 0.24 | 0.26 | -0.05 |
| non-smoker | 0.60 | 0.58 | 0.03 |
| missing data | 0.16 | 0.16 | 0.02 |
| **Ambulance** | 0.29 | 0.28 | 0.02 |
| **CCI median** | 3.10 | 2.99 | 0.06 |
| **Comorbidities** | | | |
| Myocardial infarction | 0.00 | 0.00 | -0.01 |
| Congestive heart failure | 0.01 | 0.01 | 0.02 |
| Peripheral vascular disease | 0.00 | 0.00 | 0.02 |
| Cerebrovascular disease | 0.01 | 0.00 | 0.02 |
| Dementia | 0.01 | 0.00 | 0.05 |
| Chronic pulmonary disease | 0.00 | 0.00 | 0.03 |
| Peptic ulcer | 0.00 | 0.00 | 0.01 |
| Malignancy | 0.01 | 0.01 | 0.01 |
| Metastatic solid tumor | 0.00 | 0.00 | 0.04 |
| Mild liver disease | 0.00 | 0.00 | 0.02 |
| Moderate or Severe liver disease | 0.00 | 0.00 | 0.02 |
| Hemiplegia or paraplegia | 0.00 | 0.00 | 0.01 |
| Diabetes with chronic complications | 0.01 | 0.01 | 0.03 |
| Diabetes without chronic complications | 0.01 | 0.01 | 0.03 |
| Rheumatic disease | 0.00 | 0.00 | 0.01 |
| **Consciousness levels,** | | | |
| 0 (Alert) | 0.84 | 0.85 | -0.03 |
| 1–3 (Delirium) | 0.13 | 0.12 | 0.03 |
| 10–30 (Somnolence) | 0.02 | 0.02 | 0.00 |
| 100–300 (Coma) | 0.01 | 0.01 | -0.03 |
| **Interventions within 2 days of admission** | | | |
| Anticoagulation | 0.04 | 0.02 | 0.08 |
| Inhaled ciclesonide | 0.00 | 0.00 | 0.00 |
| Molnupiravir | 0.01 | 0.00 | 0.01 |
| Remdesivir | 0.06 | 0.05 | 0.03 |

Abbreviations: IPTW, inverse probability of treatment weighting; BMI, body mass index; CCI, Charlson Comorbidity index.

**Table 3. The outcome measures before and after propensity score matching.**

| | Unmatched groups | | | Propensity score-matched groups | | |
|---|---|---|---|---|---|---|
| | Antibiotic prescribing group | Control group | p | Antibiotic prescribing group | Control group | P |
| Outcome, *n* (%) | n = 3,873 | n = 140,237 | | n = 3,861 | n = 3,861 | |
| 28-day mortality | 91 (2.4) | 4,291 (3.1) | 0.01 | 90 (2.3) | 140 (3.6) | <0.01 |
| In-hospital mortality | 154 (4.0) | 5,935 (4.2) | 0.47 | 153 (4.0) | 193 (5.0) | 0.03 |
| CDI | 22 (0.6) | 19 (0.01) | <0.01 | 23 (0.6) | 3 (0.1) | <0.01 |
| AKI | 14 (0.4) | 39 (0.03) | 0.23 | 14 (0.4) | 6 (0.2) | 0.12 |

*Abbreviations:* CDI, *Clostridioides difficile* infection; AKI, acute kidney injury.

**Table 4. The outcome measures in IPTW analysis.**

| IPTW analysis group | |
|---|---|
| Outcomes | Estimated value (95% Confidence intervals) |
| 28-day mortality | -0.70 (-0.98 to -0.42) |
| In-hospital mortality | -0.44 (-0.65 to -0.23) |
| CDI | 0.39 (0.08 to 0.70) |
| AKI | 0.15 (-0.12 to 0.42) |

*Abbreviations:* IPTW, inverse probability of treatment weighting; BMI, body mass index; CCI, Charlson Comorbidity index.

**Table 5. Impact of antibiotic prescribing on outcomes using GEEs.**

| Outcomes | Odds ratio (95% Confidence intervals) |
|---|---|
| 28-day mortality | 0.7 (0.5–0.8) |
| In-hospital mortality | 0.8 (0.7–0.95) |
| CDI | 4.7 (2.5–8.6) |
| AKI | 1.4 (0.8–2.7) |

*Abbreviations:* CDI; *Clostridioides difficile* infection, AKI; acute kidney injury.

## Discussion

Our findings showed that antibiotic prescriptions were associated with a reduction in both the 28-day and in-hospital mortalities among non-critically ill hospitalized patients with COVID-19. Conversely, antibiotic use was associated with an increase in CDI, but no significant differences were observed in AKI.

The WHO reports that during the COVID-19 pandemic, approximately 75% of COVID-19 patients received unnecessary 'just in case' antibiotic prescriptions. It has been suggested that these prescriptions do not improve clinical outcomes for COVID-19 patients and may even be harmful [24].

In this study, antibiotic prescriptions led to a reduction in mortality. Multiple reasons exist why antibiotic usage decreases mortality, resulting in lower 28-day and in-hospital mortalities compared with the control group. Bacterial bronchiolitis may be present in patients with mild COVID-19. In contrast, bacterial coinfections and secondary bacterial pneumonia may occur

in such patients. Patients with chronic lung diseases, such as chronic obstructive pulmonary disease, or older adults are at high risk for bacterial bronchitis and pneumonia [25]. Hence, antibiotics are needed for these populations. Antibiotic use benefits may be observed in high-risk patients [26,27].

Furthermore, previous studies showed that co-infection with SARS-CoV-2 and bacteria was observed in 3.5% of cases, followed by secondary bacterial infections in 14.3% of cases [13]. In Omicron variants, secondary bacterial infections were reported to occur in 10.1% of cases [28]. Additionally, coinfections and secondary bacterial infections were observed at consistent rates in all patients with COVID-19 [29–32]. Therefore, antibiotic use within 2 days of admission may affect these bacterial infections, reducing the 28-day and in-hospital mortality rates.

CDI incidence is 1.3%–1.4% in the general population [33] and 0.6% in the United States [34], similar to our results. Although antibiotic use was associated with increased CDI incidence, its prevalence was lower compared with previous studies. The strict infection control measures implemented during the COVID-19 pandemic may have effectively reduced the incidence and transmission of CDI.

Our study revealed that antibiotic use was not associated with an increase in AKI (per our definition). AKI mechanisms in patients with COVID-19 may include direct viral injury and systemic inflammatory responses, such as sepsis [35,36]. Studies have reported an AKI incidence of approximately 30% in patients with COVID-19, increasing to 50% in critically ill patients [37,38]. In a previous study, most AKI cases in patients with COVID-19 were mild or moderate, not requiring renal replacement therapy [39,40]. Our study did not include laboratory data or urine volumes, which may have introduced bias into our results.

## Strengths and limitations

The strengths of our study are: 1) the results reflect real-world clinical settings and treatment strategies for COVID-19 in Japan; and 2) our study includes the largest number of non-critically ill patients with COVID-19 for investigating the effect of antibiotics compared with previous studies.

However, our study has some limitations. First, it was a retrospective, multicenter database study, and confounding factors may have influenced our results. Second, our database did not include laboratory data, vital signs, computed tomography findings, or detailed clinical courses. Third, the cause of death was uncertain, as mortality rates in this study include both COVID-19-related and other causes. Fourth, the reasons for antibiotic prescriptions are unknown. Fifth, the database did not include information on SARS-CoV-2 vaccinations. Sixth, the database lacked information regarding the number of antibiotics used. Finally, we used propensity score matching analysis to adjust for patient background characteristics, achieving an adequate balance between the groups. However, confounding factors from unmeasured covariates cannot be completely avoided; therefore, bias cannot be ruled out.

## Conclusion

This study, using the DPC database, showed that antibiotic prescriptions within 2 days of admission reduced both 28-day and in-hospital mortality rates in non-critically ill hospitalized patients with COVID-19. However, antibiotic prescriptions within 2 days of admission increased the risk of CDI. Conversely, antibiotic prescriptions within 2 days of admission did not increase the risk of AKI.

## Acknowledgments

We would like to thank Editage for the English language editing.

## Author contributions

**Conceptualization:** Eiichi N. Kodama.

**Data curation:** Haruka Imai, Tomoki Mizuno, Shota Takahashi, Hideya Itagaki, Makiko Yoshida.

**Formal analysis:** Haruka Imai, Jun Suzuki.

**Funding acquisition:** Shiro Endo, Eiichi N. Kodama.

**Investigation:** Haruka Imai.

**Methodology:** Haruka Imai.

**Project administration:** Shiro Endo.

**Supervision:** Shiro Endo, Eiichi N. Kodama.

**Validation:** Haruka Imai.

**Writing – original draft:** Haruka Imai.

**Writing – review & editing:** Eiichi N. Kodama.

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
