## [Decision Letter · Decision Letter 0]

4 Dec 2024

PONE-D-24-46598The effect of antibiotic prescription in non-critically ill hospitalized patients with COVID-19: A Japanese inpatient database study

PLOS ONE

Dear Dr. Kodama,

Thank you for submitting your manuscript to PLOS ONE. After careful consideration, we feel that it has merit but does not fully meet PLOS ONE’s publication criteria as it currently stands. Therefore, we invite you to submit a revised version of the manuscript that addresses the points raised during the review process.

We look forward to receiving your revised manuscript.

Kind regards,

Benjamin M. Liu, MBBS, PhD, D(ABMM), MB(ASCP)

Academic Editor

PLOS ONE

Journal Requirements:

3. Thank you for stating the following financial disclosure: “This work was supported by JSPS KAKENHI (grant number: 23K16288) and AMED-CREST (JP22gm1610007).”

4. In this instance it seems there may be acceptable restrictions in place that prevent the public sharing of your minimal data. However, in line with our goal of ensuring long-term data availability to all interested researchers, PLOS’ Data Policy states that authors cannot be the sole named individuals responsible for ensuring data access (http://journals.plos.org/plosone/s/data-availability#loc-acceptable-data-sharing-methods). Data requests to a non-author institutional point of contact, such as a data access or ethics committee, helps guarantee long term stability and availability of data. Providing interested researchers with a durable point of contact ensures data will be accessible even if an author changes email addresses, institutions, or becomes unavailable to answer requests. Before we proceed with your manuscript, please also provide non-author contact information (phone/email/hyperlink) for a data access committee, ethics committee, or other institutional body to which data requests may be sent. If no institutional body is available to respond to requests for your minimal data, please consider if there any institutional representatives who did not collaborate in the study, and are not listed as authors on the manuscript, who would be able to hold the data and respond to external requests for data access? If so, please provide their contact information (i.e., email address). Please also provide details on how you will ensure persistent or long-term data storage and availability.

Additional Editor Comments:

Editor's comments:

Line 58-60: "Coronavirus disease 2019 (COVID-19), caused by severe acute respiratory syndrome coronavirus 2 (SARS-CoV-2), can progress from mild symptoms to acute respiratory distress syndrome with high mortality rate, hence a global pandemic": More references need to be cited, with these ones (PMID: 33337932 and 33667962) as examples (citing is optional).

line 212, Discussion: "Antibiotic use benefits may be observed in high-risk patients.": More references need to be cited, with this one (PMID: 34406882) as an example (citing is optional).

Reviewers' comments:

Reviewer's Responses to Questions

**Comments to the Author**

1. Is the manuscript technically sound, and do the data support the conclusions?

Reviewer #1: Yes

Reviewer #2: Yes

2. Has the statistical analysis been performed appropriately and rigorously? 

Reviewer #1: Yes

Reviewer #2: Yes

3. Have the authors made all data underlying the findings in their manuscript fully available?

Reviewer #1: Yes

Reviewer #2: Yes

4. Is the manuscript presented in an intelligible fashion and written in standard English?

Reviewer #1: Yes

Reviewer #2: Yes

5. Review Comments to the Author

Reviewer #1: Real-world retrospective analyses benefited from the compilation of previous data from databases. The paper evaluated the impact of antibiotic prescribing on COVID-19 non-critically ill patients by analyzing the Japanese inpatient database, including analysis of mortality, secondary nosocomial infections, and hemodialysis, which provides a valuable reference value for clinicians or researchers, in terms of treatment and prognosis.

The paper itself is logically clear, with concise arguments and clear statistics. However, even if anonymized data is purchased, please provide proof of the appropriate ethical license for this study as required by the journal.

Reviewer #2: This study aimed to evaluate the effects of antibiotic prescriptions in non-critically ill COVID-19 patients using a Japanese multicenter inpatient database.

While antibiotics help manage secondary bacterial infections, they also come with risks. The first thing that comes to mind is that antibiotic prescriptions are known to cause diarrhea as a side effect.

In this study, the authors used Clostridioides difficile infection (CDI) and acute kidney injury (AKI) as measures of the side effects of antibiotic usage, while 28-day mortality and in-hospital mortality were assessed as beneficial outcomes.

the definition of "mortality" in this study is unclear. Is it attributed to COVID-19 or other causes? The authors should clarify how they distinguish between COVID-related and unrelated mortality to improve the interpretability of their results

Introduction :The Introduction does not provide enough background information. While I didn’t search extensively, I believe there are existing papers or case reports discussing the benefits of antibiotic prescriptions for COVID-19 patients. Given that excessive antibiotic use is becoming a significant issue, I recommend the authors include more context on the benefits and side effects of antibiotic prescriptions in this population.

Methods: The terms Clostridioides difficile infection (CDI) and acute kidney injury (AKI) need clearer definitions in Lane 130.

Results: The baseline characteristics of the participants appear balanced, which is a strength of the study.

Discussion: The discussion section should compare the results with other available data. For instance, the WHO has stated that "overall, antibiotic use did not improve clinical outcomes for patients with COVID-19 and may cause harm to individuals without bacterial infections compared to those not receiving antibiotics." Including such comparisons would contextualize the findings and add depth to the discussion.

First thing jump to my head is that The side effect of antibiotic prescription is reported to casuse diarrhea. Ref: Early viral versus late antibiotic-associated diarrhea in novel coronavirus infection. The good part is antibiotics deal with secondary bacteria infection.

In this study, the author used CDI and AKI as measures of side effect of antibiotic usage, and use 28-day mortality rate and In-hospital mortality as the beneficial side outcomes.

In term of “mortality”, it is not well defined in this study. Is it from covid or others? And how to distinguish one from another.

Introduction doesn’t provide enough information/background: to be honsst, I didn’t search carefully but. I believe there should be some papers/case reports talking about the benefitcial of antibiotic prescribing to patients. becasue Now the excessive usega of antibiotics is becoming a problem. In this case, I would suggest authors to : Add some context about the benefits or side effect of antibiotix prescription on covid patients .

Definition of CDI and AKI: Clostridioides difficile infection (CDI) and acute kidney injury (AKI). Lane 130 need a clearer definition.

Result: the Baseline characteristics of the participants are balanced

Discussion section shold also discuss the result comparing with other available data. For example,WHO announced: Overall, antibiotic use did not improve clinical outcomes for patients with COVID-19 and also . But rather, it might create harm for people without bacterial infection, compared to those not receiving antibiotics’

6. PLOS authors have the option to publish the peer review history of their article (what does this mean? ). If published, this will include your full peer review and any attached files.

**Do you want your identity to be public for this peer review?** For information about this choice, including consent withdrawal, please see our Privacy Policy .

Reviewer #1: No

Reviewer #2: No

---

## [Author Response · Author response to Decision Letter 0]

17 Jan 2025

Thank you for your feedback. We will respond to all comments.

We have confirmed that the manuscript conforms with the journal submission rules. We have italicized the title (lines 3, 4, and 6).

We confirmed that the funding was mentioned not in the manuscript itself, but in the funding statement.

3. Thank you for stating the following financial disclosure: “This work was supported by JSPS KAKENHI (grant number: 23K16288) and AMED-CREST (JP22gm1610007).”

Thank you for your advice. Jun Suzuki and Haruka Imai received support from JSPS-KAKENHI (grant no. 23K16288). Eiichi Kodama was supported by AMED-CREST (grant no. JP22gm1610007). Neither JSPS-KAKENHI (grant no. 23K16288) nor AMED-CREST (JP22gm1610007) were involved in this work, and the recommended text has been added in lines 322–324.

「The funders had no role in study design, data collection and analysis, decision to publish, or preparation of the manuscript.」

4. In this instance it seems there may be acceptable restrictions in place that prevent the public sharing of your minimal data. However, in line with our goal of ensuring long-term data availability to all interested researchers, PLOS’ Data Policy states that authors cannot be the sole named individuals responsible for ensuring data access (http://journals.plos.org/plosone/s/data-availability#loc-acceptable-data-sharing-methods). Data requests to a non-author institutional point of contact, such as a data access or ethics committee, helps guarantee long term stability and availability of data. Providing interested researchers with a durable point of contact ensures data will be accessible even if an author changes email addresses, institutions, or becomes unavailable to answer requests. Before we proceed with your manuscript, please also provide non-author contact information (phone/email/hyperlink) for a data access committee, ethics committee, or other institutional body to which data requests may be sent. If no institutional body is available to respond to requests for your minimal data, please consider if there any institutional representatives who did not collaborate in the study, and are not listed as authors on the manuscript, who would be able to hold the data and respond to external requests for data access? If so, please provide their contact information (i.e., email address). Please also provide details on how you will ensure persistent or long-term data storage and availability.

Thank you for your advice. The MDV company manages the raw data. You can contact the MDV company through their website here: https://www.mdv.co.jp/contactus/.

Thank you for your advice. IRB information has been moved to the Methods section (lines 183–186).

Thank you for your advice. I have checked all the references again.

Additional Editor Comments:

Editor's comments:

Line 58-60: "Coronavirus disease 2019 (COVID-19), caused by severe acute respiratory syndrome coronavirus 2 (SARS-CoV-2), can progress from mild symptoms to acute respiratory distress syndrome with high mortality rate, hence a global pandemic": More references need to be cited, with these ones (PMID: 33337932 and 33667962) as examples (citing is optional).

Thank you for your advice. We have incorporated the suggested references (PMID: 33337932 and 33667962) to support the statements in line 69. Additionally, we have added a new reference（PMID: 37629267） on line 69 to further substantiate our discussion.

line 212, Discussion: "Antibiotic use benefits may be observed in high-risk patients.": More references need to be cited, with this one (PMID: 34406882) as an example (citing is optional).

Thank you for your advice. We have incorporated the suggested reference (PMID: 34406882) to support the statement on line 263 regarding the benefits of antibiotic use in high-risk patients. Additionally, we have added a new reference on line 263 to further substantiate our discussion.

Reviewer #1: Real-world retrospective analyses benefited from the compilation of previous data from databases. The paper evaluated the impact of antibiotic prescribing on COVID-19 non-critically ill patients by analyzing the Japanese inpatient database, including analysis of mortality, secondary nosocomial infections, and hemodialysis, which provides a valuable reference value for clinicians or researchers, in terms of treatment and prognosis.

The paper itself is logically clear, with concise arguments and clear statistics. However, even if anonymized data is purchased, please provide proof of the appropriate ethical license for this study as required by the journal.

We have addressed your concern regarding the ethical license for our study. After contacting MDV, we reviewed their ethics guidelines as recommended. To ensure compliance with the journal's requirements, we have included the following statement in our manuscript (lines 88–93):

「MDV employs anonymized processed data (e.g., medical records and test results) provided by medical institutions for analysis and service development, ensuring strict confidentiality. During processing, personally identifiable information, such as date of birth and address, is obscured to prevent patient identification. This anonymized data is securely provided, for instance, by being downloaded from our servers or uploaded to the designated server [18]. 」

Reviewer #2: This study aimed to evaluate the effects of antibiotic prescriptions in non-critically ill COVID-19 patients using a Japanese multicenter inpatient database.

While antibiotics help manage secondary bacterial infections, they also come with risks. The first thing that comes to mind is that antibiotic prescriptions are known to cause diarrhea as a side effect.

In this study, the authors used Clostridioides difficile infection (CDI) and acute kidney injury (AKI) as measures of the side effects of antibiotic usage, while 28-day mortality and in-hospital mortality were assessed as beneficial outcomes.

the definition of "mortality" in this study is unclear. Is it attributed to COVID-19 or other causes? The authors should clarify how they distinguish between COVID-related and unrelated mortality to improve the interpretability of their results

We appreciate your invaluable advice. We have carefully considered your comments and made the necessary revisions in the relevant sections of the manuscript. Please find our detailed responses below:

Introduction :The Introduction does not provide enough background information. While I didn’t search extensively, I believe there are existing papers or case reports discussing the benefits of antibiotic prescriptions for COVID-19 patients. Given that excessive antibiotic use is becoming a significant issue, I recommend the authors include more context on the benefits and side effects of antibiotic prescriptions in this population.

Methods: The terms Clostridioides difficile infection (CDI) and acute kidney injury (AKI) need clearer definitions in Lane 130.

We appreciate your invaluable advice. We have carefully considered your comments and made the necessary revisions in the relevant sections of the manuscript. Please find our detailed responses below:

Results: The baseline characteristics of the participants appear balanced, which is a strength of the study.

Discussion: The discussion section should compare the results with other available data. For instance, the WHO has stated that "overall, antibiotic use did not improve clinical outcomes for patients with COVID-19 and may cause harm to individuals without bacterial infections compared to those not receiving antibiotics." Including such comparisons would contextualize the findings and add depth to the discussion.

First thing jump to my head is that The side effect of antibiotic prescription is reported to casuse diarrhea. Ref: Early viral versus late antibiotic-associated diarrhea in novel coronavirus infection. The good part is antibiotics deal with secondary bacteria infection.

In this study, the author used CDI and AKI as measures of side effect of antibiotic usage, and use 28-day mortality rate and In-hospital mortality as the beneficial side outcomes.

In term of “mortality”, it is not well defined in this study. Is it from covid or others? And how to distinguish one from another.

The database contains deaths related to and unrelated to COVID-19. The following text has been added as a limitation (lines 293–294):

「Third, the cause of death was uncertain. The mortality rates in this study include both COVID-19-related and other causes.」

Introduction doesn’t provide enough information/background: to be honsst, I didn’t search carefully but. I believe there should be some papers/case reports talking about the benefitcial of antibiotic prescribing to patients. becasue Now the excessive usega of antibiotics is becoming a problem. In this case, I would suggest authors to : Add some context about the benefits or side effect of antibiotix prescription on covid patients .

Thanks for your advice. We have added a paragraph about the benefits and risks of antimicrobial treatments for COVID-19 patients (lines 71–73):

「A possible benefit of antimicrobial use in COVID-19 patients is the possibility of reduced mortality. Disadvantages include increased antimicrobial resistance, diarrhea, other gastrointestinal problems, and Clostridioides difficile infection.」

Definition of CDI and AKI: Clostridioides difficile infection (CDI) and acute kidney injury (AKI). Lane 130 need a clearer definition.

Thank you for your feedback. We attempted to collect CDI and AKI data based on ICD-10 codes, but these were infrequently included. As laboratory and bacterial culture results were not available, we used definitions based on therapeutic agents. The following statement has been added to lines 145–147:

「We attempted to detect CDI and AKI using ICD-10 disease codes, which were rarely recorded in this database. Definitions were based on therapeutic agents, as the database lacks laboratory data and bacterial culture results.」

Result: the Baseline characteristics of the participants are balanced

Discussion section shold also discuss the result comparing with other available data. For example,WHO announced: Overall, antibiotic use did not improve clinical outcomes for patients with COVID-19 and also . But rather, it might create harm for people without bacterial infection, compared to those not receiving antibiotics’

Thank you for your suggestion. We have incorporated the WHO's recommendations into lines 252–255 as follows:

「The WHO reports that during the COVID-19 pandemic, approximately 75% of COVID-19 patients received unnecessary 'just in case' antibiotic prescriptions. It has been suggested that these prescriptions do not improve clinical outcomes for COVID-19 patients and may even be harmful [24].

---

## [Editor Report · Decision Letter 1]

22 Jan 2025

The effect of antibiotic prescription in non-critically ill hospitalized patients with COVID-19: A Japanese inpatient database study

PONE-D-24-46598R1

Dear Dr. Kodama,

We’re pleased to inform you that your manuscript has been judged scientifically suitable for publication and will be formally accepted for publication once it meets all outstanding technical requirements.

Kind regards,

Benjamin M. Liu, MBBS, PhD, D(ABMM), MB(ASCP)

Academic Editor

PLOS ONE
---

## [Editor Report · Acceptance letter]

PONE-D-24-46598R1

PLOS ONE

Dear Dr. Kodama,

I'm pleased to inform you that your manuscript has been deemed suitable for publication in PLOS ONE. Congratulations! Your manuscript is now being handed over to our production team.

Kind regards,

on behalf of

Dr. Benjamin M. Liu

Academic Editor

PLOS ONE